# OpenReview forum: "Effective and Efficient Jailbreaks of Black-Box LLMs with Cross-Behavior Attacks"
_ICLR.cc/2026/Conference — Submitted to ICLR 2026_

### Official Review · Reviewer_mLEf · 2025-10-15

**Soundness:** 3
**Presentation:** 3
**Contribution:** 3
**Rating:** 2
**Confidence:** 5

**Summary:**

The paper introduces **JCB**, a black-box *cross-behavior* jailbreak method that automatically generates and perturbs seed prompts (using part-of-speech preserving synonym substitution) and leverages a scorer LLM to reuse successful prompts across behaviors, enabling efficient, automated discovery of jailbreaks.
Empirically, JCB attains substantially higher attack success while using far fewer queries (up to 94% fewer) than prior black-box baselines, shows strong zero-shot transferability across LLMs, and includes analyses of efficiency, transferability, and comparison with high-complexity methods like AutoDAN-Turbo. The authors highlight JCB’s practicality as a low-complexity red-teaming tool but note limitations (dataset/metric dependence and remaining headroom in ASR), which they discuss and position as future work.

**Strengths:**

- **Originality & significance:** JCB introduces a simple, black-box cross-behavior jailbreak technique that pairs POS-preserving synonym substitution with a scorer LLM to recycle effective seeds, substantially lowering query costs and improving automated red-teaming and cross-model transfer.
- **Quality & clarity**: The paper supplies comprehensive empirical evidence (efficiency, transferability, and baseline comparisons), clear methodological and prompt engineering details, and an honest limitations section, supporting reproducibility and practical impact.
- **New methods**: They propose JCB, a novel, effective, efficient, and low-cost approach that leverages cross-behavior attacks to autonomously discover jailbreak prompts without human intervention

**Weaknesses:**

- The authors do not include a reproducibility statement. I am willing to increase the score if you provide a reproducibility statement and extend the experiments.

- The paper’s novelty is limited: its theoretical mechanism closely mirrors LLMFuzzer—while LLMFuzzer relies on human-provided seed files, the automatic seed-generation here appears to be a relatively simple, incremental change.

- **Incomplete evaluation:** The study omits recent SOTA attack baselines (e.g.,  BOOST, GPTFuzzer, FlipAttack).

**Questions:**

- How does this paper’s novelty concretely differ from LLM-Fuzzer—i.e., what new mechanism, theoretical claim, or empirical advantage does it introduce beyond mutation-based fuzzing?

- What is the method's performance under SOTA defense methods, such as self-reminder

---

> ### Author Response · Authors · 2025-11-22
> **Authors’ Response to Reviewer mLEf**
>
> We sincerely thank the reviewer for appreciating the originality and practicality of our method, the quality and clarity of the details, and the extensive empirical evaluation. We now address all concerns.
>
> &nbsp;
>
> ---
>
> ### **Q1. Missing reproducibility statement**
>
> Thank you for pointing this out. We apologize for missing this in the original submission. In the revised manuscript, we have added a dedicated Reproducibility section (Section 7) that clearly specifies our datasets, prompts, hyperparameters, and evaluation procedures, thereby allowing an independent reproduction of our results. At this stage, we are unable to release the full codebase because it contains personal identifiers. However, we commit to releasing a public, fully documented version of the code, along with configuration files and scripts to reproduce all main results, upon acceptance and prior to the camera-ready deadline.
>
> &nbsp;
>
> ---
>
> ### **Q2. Novelty relative to LLMFuzzer**
>
> We agree that JCB is similar to LLMFuzzer at a high level (they are both optimization based methods that rely on seed prompts), but it has two key distinguishing features:
>
> 1. JCB introduces cross-behavior knowledge transfer, which LLMFuzzer does not use. This cross-behavior learning allows reuse of successful prompts for one harmful request to other harmful requests and results in an extremely high query efficiency.
> 2. JCB’s part-of-speech-preserving synonym substitution maintains semantic validity without relying on expensive calls to the LLMs, unlike LLMFuzzer which relies on a “mutator LLM” with hand-crafted instructions.
>
> Owing to these features, JCB significantly outperforms LLMFuzzer (more than 60 percentage point improvement in ASR), as evidenced by our new experiments in Table 11, Appendix I.
>
> This new Table 11 from Appendix I is copied here for convenience:
> | Methods ↓ / Models → | GPT-3.5 Turbo 1106 | GPT-4o 2024-05-13 | GPT-4o-mini | Llama 3.1 405B | Avg.  |
> |----------------------|--------------------|-------------------|-------------|-----------------|-------|
> | PAIR                 | 70                 | 44                | 04          | 06              | 31.0  |
> | TAP                  | 64                 | 60                | 10          | 04              | 34.5  |
> | BOOST                | 22                 | 00                | 04          | 04              | 7.5   |
> | LLMFuzzer            | 26                 | 70                | 34          | 00              | 32.5  |
> | FlipAttack           | 90                 | 90                | 56          | 24              | 65.0  |
> | **JCB (Ours)**       | **100**            | **96**            | **98**      | **96**          | **97.5** |
>
>
>
>
> &nbsp;
>
> ---
>
> ### **Q3. Missing SOTA baselines (BOOST, LLMFuzzer, FlipAttack)**
>
> We have added these comparisons in the revised manuscript (Table 11, Appendix I, copied above).
>
> JCB consistently outperforms BOOST, LLMFuzzer, and FlipAttack by large margins on GPT-3.5-Turbo, GPT-4o, GPT-4o-mini, and Llama 3.1 405B.
>
> &nbsp;
>
> ---
>
> ### **Q4. Performance under SOTA defenses**
>
> Thank you for the suggestion. We ran the requested experiment and found that Self-Reminder reduces JCB’s ASR to near 0%. This is expected, as Self-Reminder is a system-level defense relying on privileged system prompting, which lies outside the user-level threat model our attack targets. We also note that such defenses are not used in frontier models, consistent with our high success rates on GPT- and Llama-family systems. This may be due to Self-Reminder’s aggressive safety filtering, which changes normal model behavior. Nonetheless, including this result improves the completeness of our evaluation, and we have added it to the revised manuscript (Appendix O). Importantly, perplexity-based filtering, a lightweight and commonly proposed defense, remains fully bypassed by our method (Figure 5, Appendix O).

---

> > ### Comment · Reviewer_mLEf · 2025-11-22
> >
> > Thanks author response and provide a huge of experiments, but the novelty is still a huge concern. I update my score and keep my rating as weak reject.

---

> > > ### Author Response · Authors · 2025-11-22
> > > **Follow-up about our work’s novelty**
> > >
> > > Dear Reviewer,
> > >
> > >
> > > Thank you for your prompt follow-up and for the positive feedback on our revision. We appreciate you taking the time to re-evaluate our work.
> > >
> > > Regarding the concern about novelty, we would like to highlight two key experimental additions in the revised manuscript:
> > >
> > > 1. Our ablation study (Table 12, Appendix K) shows that both cross-behavior learning mechanism and the lightweight synonym-based perturbation component are essential to JCB’s effectiveness. Removing them leads to substantial drops in attack success, demonstrating that JCB's gains cannot be attributed to simple mutation-based fuzzing alone.
> > >
> > > The new Table 12 from Appendix K is copied here for convenience:
> > > | Config. | All Themes Seeds | “Good Themes” Seeds | “Good Themes” Seeds + *Selection* | “Good Themes” Seeds + *Selection* + *Perturbation* |
> > > |---------|------------------|---------------------|------------------------------------|----------------------------------------------------|
> > > | ASR     | 59.00            | 66.00               | 70.50                              | **76.75**                                          |
> > >
> > >
> > > 2. We additionally include an analysis of the underlying mechanism behind JCB’s effectiveness (Section L). This analysis shows that JCB consistently pushes harmful prompts away from harmful latent regions and toward harmless regions in the model’s internal representation space, effectively bypassing the refusal boundary. This latent-space explanation is, to our knowledge, absent in prior fuzzing-style methods such as LLMFuzzer.
> > >
> > >
> > > Finally, we genuinely appreciate your time and your feedback. If there are additional analyses or experimental comparisons that you believe would help convince you of our work’s novelty, we would be very happy to incorporate them. Please do not hesitate to let us know; your guidance is very valuable to us.

---

> > > > ### Comment · Reviewer_mLEf · 2025-11-22
> > > >
> > > > Thanks for your response and I will look at the other reviewer's response later also. If I have any comments, I will let you know. And I pretty thanks your revision work on addtional experiment. However, I need to mention to you one mistake. Point 2 is already being claimed by the BOOST paper, so it is not your contribution. I can only regard the Figure 4 right part as your contribution and remember to citi the BOOST paper as the aligned observation in the BOOST paper.

---

> > > > > ### Author Response · Authors · 2025-11-26
> > > > > **Clarification and new results against Self-Reminder**
> > > > >
> > > > > Dear Reviewer,
> > > > >
> > > > > We apologize for the confusion regarding point 2 above. We agree that BOOST presents an analysis of its jailbreak mechanism and we have cited it in Appendix L in the revised manuscript. However, we included point 2 above since the review question asked about differences of our method from LLMFuzzer and we believed it was important to highlight the difference. Unlike JCB, LLMFuzzer does not provide any latent-space explanation about the reasons for its performance. Regardless of this, we thank you for being open about considering the right part of Figure 4 as our contribution.
> > > > >
> > > > > Additionally, to address another reviewer’s (Reviewer Se2F’s) concern about JCB’s effectiveness against the Self-Reminder defense, we conducted new experiments in an **adaptive setting**, where the attacker is aware of the defense and instructs the model to operate without Self-Reminder’s restrictions. Under this setting, JCB successfully bypasses the defense, achieving an average ASR of 99.5% across all evaluated models (Table 15, Appendix O in the revised manuscript).
> > > > >
> > > > > This new Table 15 is copied here for convenience:
> > > > > | Models | GPT-3.5 Turbo 1106 | GPT-4o 2024-05-13 | GPT-4o-mini | Llama 3.1 405B | Avg.  |
> > > > > |----------------------|--------------------|-------------------|-------------|-----------------|-------|
> > > > > | JCB ASR          | 100                 | 100                | 98          | 100              | 99.5  |
> > > > >
> > > > >
> > > > > We are informing you here since this is relevant to “Q4. Performance under SOTA defenses” in our first response to your review above.

---

> > > > > > ### Comment · Reviewer_mLEf · 2025-11-26
> > > > > >
> > > > > > Thanks author further response and I keep my score currently.  I will not strongly claim reject if other reviewers agree to accept.

---

### Official Review · Reviewer_Se2F · 2025-10-26

**Soundness:** 2
**Presentation:** 3
**Contribution:** 2
**Rating:** 4
**Confidence:** 3

**Summary:**

This paper proposes a prompt-framework generation method for black-box model jailbreak attacks. The authors identify three main shortcomings of existing automated red-team/jailbreak approaches: low attack efficiency (requiring many queries); reliance on expensive auxiliary LLMs or white-box gradient information; and generated prompts that are often semantically distorted or easily detected by filters. To address these issues, the paper introduces a cross-behavior black-box jailbreak method, JCB, whose core idea is to transfer successful experiences across different harmful behaviors to improve the efficiency of jailbreaking new behaviors. Experiments on 20+ open-source and closed-source models demonstrate that the method effectively increases jailbreak success rates.

**Strengths:**

1. The paper conducts experiments on a large number of open-source and closed-source large models; the reported attack success rates are clearly better than existing methods, demonstrating empirical advantages for the proposed approach.
2. The paper is well-structured and easy to read.

**Weaknesses:**

1. **Unverified claims about limitations.** The paper asserts that existing methods generate prompts that are often semantically distorted or easily detected by filters, but it does not validate this claim against state-of-the-art jailbreak defenses. Experiments are only conducted on aligned models without defenses.
2. **Weak theoretical contribution.** The paper is primarily motivated by the observation that successfully used jailbreak prompt templates can be reused for more malicious behaviors, but it does not investigate the underlying mechanisms. For example, when the same template is applied to different malicious behaviors, what trends appear in the model's feature space that determine the strength of transferability? Given the large body of empirical work on jailbreak attacks and defenses, deeper investigation into the principles behind different methods would likely be more valuable.
3. **Unfair/insufficient experimental setup and lack of necessary ablations.** The paper claims the proposed method requires no manual guidance or jailbreak knowledge, yet it relies on an important prior: the attacker must generate a certain number of successful jailbreak prompt frameworks, which evidently depend on manual guidance or jailbreak expertise. The paper also lacks appropriate ablation studies to verify the effect of these priors on the method’s effectiveness—for example, how the attack success rate of the initial jailbreak prompts influences the overall method's performance.

**Questions:**

Please refer to weaknesses.

---

> ### Author Response · Authors · 2025-11-22
> **Authors’ Response to Reviewer Se2F**
>
> We thank the reviewer for recognizing the clarity of the writing, the large-scale evaluation across many LLMs, and the strong empirical performance. We address all concerns below.
>
> &nbsp;
>
> ---
>
> ### **Q1. Claims about semantically distorted/easily detected prompts**
>
> We have now added explicit validation in the revised manuscript:
>
> * Perplexity-based defense evaluation (Appendix O): JCB prompts remain well below the detection threshold, while GCG prompts are easily flagged.
> * This directly confirms the claim that our synonym-based perturbation avoids nonsensical or statistically anomalous text.
>
> &nbsp;
>
> ---
>
> ### **Q2. Lack of investigation into underlying mechanisms**
>
> We added a new mechanistic analysis on JCB (Appendix L):
>
> * Using residual stream activations, we show that JCB pushes prompts away from the harmful region in the latent space and closer to the harmless region (Figure 4).
> * We also show (using projections on the “compliance” direction) that JCB’s successful jailbreak prompts are indeed closer to the harmless region compared to the unsuccessful jailbreak prompts.
> * This provides insight into *why* JCB prompts are successful: they shift activations into regions similar to harmless prompts, bypassing the safety boundary.
>
> &nbsp;
>
> ---
>
> ### **Q3.  Insufficient experimental setup and lack of ablation studies**
>
> We thank the reviewer for raising this point.
>
> Regarding the concern that our method “requires no manual guidance or jailbreak knowledge” yet relies on two effective jailbreak themes, we clarify that these themes are **not manually engineered by us**. Instead, they are **publicly documented findings from prior work** showing that certain high-level prompt structures tend to yield higher jailbreak success rates. Using such insights is a standard and essential part of scientific progress, and not a form of privileged domain knowledge.
>
> More importantly, our ablation study in the revised manuscript (Table 12, Appendix K) shows that while these themes improve ASR, the other components of JCB (the scorer-based weighted selection component and the part-of-speech-preserving perturbation component) contribute substantially to the success. These ablations demonstrate that the full method’s performance does not depend on any single design choice, but rather benefits from general patterns identified in the literature as well as each component of JCB.
>
> The new Table 12 from Appendix K is copied here for convenience:
> | Config. | All Themes Seeds | “Good Themes” Seeds | “Good Themes” Seeds + *Selection* | “Good Themes” Seeds + *Selection* + *Perturbation* |
> |---------|------------------|---------------------|------------------------------------|----------------------------------------------------|
> | ASR     | 59.00            | 66.00               | 70.50                              | **76.75**                                          |

---

> ### Comment · Reviewer_Se2F · 2025-11-25
>
> I have reviewed the comments from other reviewers and the authors' responses. I believe the authors have partially addressed my concerns, particularly the theoretical analysis section, which is quite interesting. However, considering that other reviewers have also pointed out issues regarding the paper's contribution, and that existing defense methods have been proven effective against such attacks, I am inclined to maintain my score.

---

> > ### Author Response · Authors · 2025-11-26
> > **Follow-up**
> >
> > Dear Reviewer,
> >
> > &nbsp;
> >
> > First, we want to thank you for appreciating the new theoretical analysis and for confirming that it addresses one of your concerns.
> >
> > &nbsp;
> >
> > Regarding the concern raised by other reviewers about the effectiveness of our method (JCB) under the Self Reminder defense: In our original response, we only reported results in the **non-adaptive setting** (i.e., where the attacker is unaware of the defense) due to resource constraints and the need to prioritize other major experiments (e.g., ablations, theoretical analysis, and broader baseline comparisons). Under that setting, as noted previously, JCB does not bypass Self Reminder.
> >
> > &nbsp;
> >
> > However, since this point is causing the reviewer to question JCB’s overall efficacy, we have now conducted new experiments in an **adaptive setting**, where the attacker is aware of the deployed defense. In this setting, JCB successfully circumvents the Self Reminder defense by simply incorporating instructions to ignore Self Reminder’s system prompt and incorporating it within JCB. Across all tested models, JCB achieves high adaptive ASR, as shown below and included in the revised manuscript (Table 15, Appendix O):
> >
> > &nbsp;
> >
> > | Models | GPT-3.5 Turbo 1106 | GPT-4o 2024-05-13 | GPT-4o-mini | Llama 3.1 405B | Avg.  |
> > |----------------------|--------------------|-------------------|-------------|-----------------|-------|
> > | JCB ASR          | 100                 | 100                | 98          | 100              | 99.5  |
> >
> > &nbsp;
> >
> > In summary, we believe our original response addressed your concerns about the semantic coherence of our attack prompts, investigation into the underlying mechanisms, and ablation studies. Additionally, we hope these new results fully address your concern about the effectiveness of our attack under the Self Reminder defense. We hope you will reconsider updating your score given the new evidence we presented above.
> >
> > &nbsp;
> >
> > Finally, we genuinely appreciate your feedback. It has really helped improve our work. If you have any additional concerns, we would be more than happy to address them.

---

### Official Review · Reviewer_DUEw · 2025-10-31

**Soundness:** 2
**Presentation:** 2
**Contribution:** 2
**Rating:** 4
**Confidence:** 4

**Summary:**

This paper proposes JCB (Jailbreak with Cross-Behavior attacks), a black-box jailbreak method designed to efficiently discover harmful prompts for LLMs. JCB leverages prior knowledge about LLMs weaknesses to generate seed prompts, then perturbates them by synonym substitution as a perturbation method. Experiments on the HarmBench and AdvBench benchmarks show that JCB achieves high ASR and requires fewer queries compared to baselines such as PAIR and TAP. The method also demonstrates strong zero-shot transferability across diverse open- and closed-source LLMs.

**Strengths:**

1. Clear motivation and practical efficiency: The paper clearly identifies the high query and cost inefficiency of prior black-box jailbreak methods and proposes a lightweight method that avoids reliance on external LLM calls. The design choice, synonym-based perturbation, make JCB simple and computationally efficient, aligning well with the stated goals.
2. Comprehensive empirical evaluation: The authors evaluate JCB on both HarmBench and AdvBench datasets, covering a wide range of open- and closed-source LLMs. Results consistently show strong ASR and query efficiency, suggesting that the method is both effective and robust across different models.

**Weaknesses:**

1. Lack of ablation studies and unclear source of performance gains: While the proposed JCB is built upon a standard black-box jailbreak optimization pipeline, the paper reports notably superior performance compared to baselines without clearly identifying the source of improvement. JCB relies on two “most successful” jailbreak themes to generate seed prompts, and this design choice seems crucial. However, no ablation study is provided to quantify the effect of theme selection. If seed prompts are so important, their intrinsic jailbreak capability should be evaluated independently. Moreover, the paper claims that synonym substitution as a perturbation method contributes to higher success rates, but it remains unclear whether the performance gains stem from synonym replacement itself or from the strong prior knowledge introduced through theme selection. Similarly, the reported query efficiency advantage may also result primarily from this prior knowledge rather than the perturbation design. A more detailed ablation analysis is needed to clarify these factors.

2. Limited diversity and novelty of discovered jailbreaks: The goal of jailbreak research is to uncover new safety weaknesses in LLMs. However, the jailbreak prompts generated by JCB appear to be confined to the two selected themes (“assumed responsibility” and “character roleplay”). As a result, JCB may not reveal fundamentally new vulnerabilities, and the diversity of generated prompts could be limited. This restricts the method’s broader contribution to improving LLM safety by identifying novel failure modes.

3. Concerns about experimental fairness: JCB leverages strong prior knowledge through theme selection, which provides it with a substantial advantage over weaker baselines such as PAIR. This raises concerns about the fairness of the comparisons. Although Section 4.6 argues that comparisons with AutoDAN-Turbo are “unfair” due to its higher complexity, JCB’s own use of domain priors arguably creates a similar imbalance when compared against other black-box baselines.

4. Concerns regarding implementation correctness and comparability: It appears that the baseline results reported in the paper are directly cited from AutoDAN-Turbo paper rather than reproduced within the same experimental setup. This makes it difficult to verify whether the evaluation pipeline is consistent across methods and raises concerns about the comparability and reliability of the reported results.

**Questions:**

1. Are all the baseline results reported in this paper directly taken from the AutoDAN-Turbo pape?

---

> ### Author Response · Authors · 2025-11-22
> **Authors’ Response to Reviewer DUEw**
>
> We thank the reviewer for highlighting the clarity of our motivation, the efficiency of our design, and the strength of our empirical evaluation. We address each weakness below.
>
> &nbsp;
>
> ---
>
> ### **Q1. Lack of ablation studies and unclear source of gains**
>
> We have added a detailed ablation study in the revised manuscript (Table 12, Appendix K) that isolates the relative contributions of the two successful jailbreak themes, the selection mechanism, and the perturbation component.
>
> This Table 12 from Appendix K is copied here for convenience:
> | Config. | All Themes Seeds | “Good Themes” Seeds | “Good Themes” Seeds + *Selection* | “Good Themes” Seeds + *Selection* + *Perturbation* |
> |---------|------------------|---------------------|------------------------------------|----------------------------------------------------|
> | ASR     | 59.00            | 66.00               | 70.50                              | **76.75**                                          |
>
> The ablations show a **consistent, stepwise improvement**, confirming that each component contributes to the final performance and that gains are not solely due to theme selection.
>
> &nbsp;
>
> ---
>
> ### **Q2. Limited diversity and novelty of discovered jailbreaks**
> We agree that relying on two high-performing jailbreak themes may lead to a narrower stylistic variety of prompts. However, this design choice is *intentionally beneficial* and empirically validated: the ablation study (Table 12, Appendix K, copied above) shows that using these two themes improves ASR compared to using all themes, confirming that they provide a good foundation for successful attacks.
>
> At the same time, even with potentially less diverse prompt structures, JCB retains substantial practical value. As shown in our main results, JCB consistently outperforms state-of-the-art baselines across a wide variety of models while requiring only a fraction of the queries. This demonstrates that high impact does not require maximal stylistic diversity: a smaller set of reliably effective patterns can still uncover vulnerabilities across many LLMs with far greater efficiency.
>
> &nbsp;
>
> ---
>
> ### **Q3. Unfair/insufficient experimental setup and reliance on priors**
>
> We appreciate the reviewer’s concern. We would like to respectfully clarify that incorporating insights from prior work, such as the identification of two particularly successful jailbreak themes, is a standard and essential scientific practice rather than an unfair advantage. Advancing research necessarily builds on established findings, and our use of these themes simply reflects leveraging well-documented observations in the literature rather than relying on any hidden or privileged knowledge. Moreover, the ablation study in the revised manuscript (Table 12, Appendix K, copied above) quantifies the effect of these themes and shows that while they improve ASR, the other components of JCB (selection and perturbation) also contribute substantially to the final performance.
>
> Regarding comparisons with AutoDAN-Turbo, we never characterize the comparison as “unfair”; instead, we highlight that AutoDAN-Turbo is query- and time-intensive, operating in a different regime that is orthogonal to our goal of an efficient yet effective red-teaming method. In fact, we explicitly acknowledge that AutoDAN-Turbo achieves higher ASR when measured solely on that metric. However, doing so requires up to two orders of magnitude more queries. For practitioners and developers seeking scalable red-teaming tools with low compute and latency requirements, JCB occupies a complementary and highly practical efficiency-oriented niche.
>
> &nbsp;
>
> ---
>
> ### **Q4. Implementation correctness concerns**
>
> Since our primary evaluation dataset is HarmBench, and because HarmBench provides a unified experimental setup and a shared implementation for all 16 baselines used in our main results (Table 1), we integrate JCB directly into that framework to ensure consistency across methods. Therefore, all baseline results reported in the main results are taken from the HarmBench codebase to ensure a fair, apples-to-apples comparison.
>
> For the auxiliary experiments such as efficiency analysis and comparison with AutoDAN-Turbo, we rely on baseline results reported in the corresponding AutoDAN-Turbo paper. This is due to the extremely high computational and monetary cost required to re-run these baselines, which require tens to hundreds of thousands of LLM queries.

---

> > ### Comment · Reviewer_DUEw · 2025-11-27
> >
> > Thank you for the authors’ response, which addressed some of my concerns. However, the main concern about the contribution of this work to improve LLMs safety is still remain, so I maintain my score.

---

> > > ### Author Response · Authors · 2025-12-02
> > > **Clarification about our work’s contribution to LLM safety**
> > >
> > > Dear Reviewer,
> > >
> > > Thank you for your follow-up and for taking the time to reassess our rebuttal. We are glad that all but one of your concerns have been addressed. We would like to offer one final clarification regarding the core contribution of our work, since the remaining concern centers on the significance of our method for improving LLM safety.
> > >
> > > A key novelty of our approach, beyond the individual components discussed earlier, is the **cross-behavior learning mechanism**, which enables JCB to *explicitly transfer* jailbreak knowledge across harmful behaviors. Instead of treating each harmful request as an isolated optimization problem, JCB learns from the structures and behavioral patterns of previously successful jailbreaks and repurposes that learned behavioral knowledge to attack for new harmful requests. To the best of our knowledge, no prior black-box jailbreak method (including TAP and PAIR) does this.
> > >
> > > This cross-behavior transfer is **not only algorithmically novel but also empirically impactful**: it is precisely this ability to accumulate and reuse behavioral knowledge that allows JCB to outperform prior state-of-the-art black-box methods by 12% ASR improvement while requiring 94% fewer queries. In contrast, existing optimization-based jailbreak methods effectively “reset” their knowledge for each target and therefore fail to benefit from the repeated structural regularities observed across harmful requests.
> > >
> > > More broadly, our work sheds light on a new direction for LLM red-teaming: one where accumulated understanding of a model’s safety weaknesses becomes an asset that can generalize across tasks. This is directly relevant for safety research because vulnerabilities that consistently transfer across behaviors represent **systematic, model-level weaknesses**, which are precisely the kinds of weaknesses safety engineers need to identify and mitigate.
> > >
> > > Even if the stylistic diversity of the resulting prompts is limited, which we agree is a reasonable point, the safety implication is substantial:
> > > * It reveals that models possess shared, cross-behavior weaknesses exploitable across diverse harmful tasks.
> > > * It shows that attackers need far less query budget than previously believed, due to the transferability of behavioral jailbreak structure.
> > > * It highlights a new threat model in which the attacker’s power compounds over time, since every successful jailbreak improves future jailbreak capability.
> > > These insights help safety evaluators understand systemic vulnerabilities, not just stylistic or prompt-pattern weaknesses.
> > >
> > > We appreciate the reviewer’s position and understand that novelty and safety impact can be subjective. However, we hope the above clarifies that JCB provides a novel algorithmic idea, a substantial empirical advancement, and a new conceptual understanding of transferable jailbreak vulnerabilities. We believe these elements together constitute a meaningful contribution to the field of LLM safety.

---

### Official Review · Reviewer_RLgU · 2025-10-31

**Soundness:** 2
**Presentation:** 2
**Contribution:** 2
**Rating:** 2
**Confidence:** 3

**Summary:**

This paper proposes JCB, a black-box jailbreak method that randomly selects seed prompts with probabilities proportional to their past attack successes, and then performs synonym-based perturbations to bypass the safety defenses of LLMs. Experimental results show that JCB achieves a high ASR while requiring relatively few queries to the target LLMs.

**Strengths:**

1. The paper provides a reasonably comprehensive set of baseline experiments.
2. JCB achieves a high ASR while requiring fewer queries to the target LLMs.

**Weaknesses:**

1. The proposed method of selecting seed prompts based on prior attack successes and performing synonym-based perturbations is quite similar to existing approaches (e.g., PAIR [1] using an LLM to rewrite prompt seeds) and thus lacks novelty.
2. Although the paper covers multiple target LLMs, the main results do not include recent state-of-the-art models. In contrast, while the later experiments evaluate newer models, they lack corresponding baseline comparisons, making it difficult to clearly assess the advantage of JCB.
3. The paper does not present the performance of JCB under mainstream defense strategies (e.g., self-reminder[2]), leaving the real-world impact and severity of the revealed vulnerabilities unclear.

---
[1] Chao P, Robey A, Dobriban E, et al. Jailbreaking black box large language models in twenty queries[C]//2025 IEEE Conference on Secure and Trustworthy Machine Learning (SaTML). IEEE, 2025: 23-42.

[2] Xie Y, Yi J, Shao J, et al. Defending chatgpt against jailbreak attack via self-reminders[J]. Nature Machine Intelligence, 2023, 5(12): 1486-1496.

**Questions:**

1. How would using different LLMs for generating the seed prompts affect the results?
2. In Line 349, “ChatGPT-4o-Latest” should specify the exact version, since using “latest” could lead to inconsistencies in future versions.

---

> ### Author Response · Authors · 2025-11-22
> **Authors’ Response to Reviewer RLgU**
>
> We sincerely thank the reviewer for recognizing the comprehensive baseline experiments and the high attack success rate with low query cost achieved by JCB. We appreciate the thoughtful feedback and address all concerns below.
>
> &nbsp;
>
> ---
>
> ### **Q1. Novelty relative to PAIR and similarity to prior methods**
>
> We agree that JCB builds on the general optimization-based jailbreak framework, but it introduces **two concrete novelties**, supported by existing as well as new experiments added in our revised manuscript:
>
> 1. Cross-behavior transfer mechanism: Unlike PAIR or TAP, JCB *reuses successful prompts across behaviors* via its scorer-based weighted selection component. The ablation study in the revised manuscript (Table 12, Appendix K) shows that this component is a key contributor to the attack performance. Additionally, this reuse of successful prompts allows JCB to be 94% more query-efficient than PAIR and TAP (Table 3, Section 4.4).
> 2. Lightweight part-of-speech-preserving synonym perturbation: Our perturbation strategy is deliberately designed to avoid semantic drift and maintain coherence, without relying on expensive calls to the “attack LLMs” used in PAIR and TAP.
>
> The new Table 12 from Appendix K is copied here for convenience:
> | Config. | All Themes Seeds | “Good Themes” Seeds | “Good Themes” Seeds + *Selection* | “Good Themes” Seeds + *Selection* + *Perturbation* |
> |---------|------------------|---------------------|------------------------------------|----------------------------------------------------|
> | ASR     | 59.00            | 66.00               | 70.50                              | **76.75**                                          |
>
>
> &nbsp;
>
> ---
>
> ### **Q2. Missing baseline comparisons for SOTA target models**
>
> We have addressed this concern in the revised manuscript:
>
> * We have added corresponding SOTA baseline comparisons against PAIR, TAP, BOOST, LLMFuzzer, and FlipAttack for large SOTA models such as GPT-4o, GPT-4o-mini, Llama 3.1 405B (Table 11, Appendix I).
> * Results show that JCB significantly outperforms all baselines on every SOTA model.
>
> The new Table 11 from Appendix I is copied here for convenience:
> | Methods ↓ / Models → | GPT-3.5 Turbo 1106 | GPT-4o 2024-05-13 | GPT-4o-mini | Llama 3.1 405B | Avg.  |
> |----------------------|--------------------|-------------------|-------------|-----------------|-------|
> | PAIR                 | 70                 | 44                | 04          | 06              | 31.0  |
> | TAP                  | 64                 | 60                | 10          | 04              | 34.5  |
> | BOOST                | 22                 | 00                | 04          | 04              | 7.5   |
> | LLMFuzzer            | 26                 | 70                | 34          | 00              | 32.5  |
> | FlipAttack           | 90                 | 90                | 56          | 24              | 65.0  |
> | **JCB (Ours)**       | **100**            | **96**            | **98**      | **96**          | **97.5** |
>
>
> &nbsp;
>
> ---
>
> ### **Q3. Lack of evaluation under defense methods**
>
> We have added a new **Performance Against Defenses** section (Appendix O). Results show:
>
> * JCB **fully bypasses** the practical perplexity-based defense.
> * JCB is unable to bypass the Self-Reminder defense under the strict threat model of non-adaptive attack (because an adaptive attacker can easily bypass the defense by modifying the reminder instructions in the system prompt). This is consistent with Self-Reminder’s status as a privileged system-level defense. However, note that such defenses are not deployed in popular frontier models, evident from our high success rates on GPT- and Llama-family systems; this may be due to Self-Reminder’s aggressive safety filtering. Nonetheless, this result clarifies the limits of our attack.
>
> &nbsp;
>
> ---
>
> ### **Q4. Effect of using different seed-prompt-generator LLMs**
>
> In the revised manuscript, the new Appendix D includes experiments studying the impact of the seed prompt generator LLM (Table 7). The performance varies substantially across LLMs (e.g., GPT-5.1 produces low-quality seeds due to strong refusals). This confirms that LLM choice affects seed quality, but also validates our decision to use ChatGPT-4o-Latest, which yields the highest ASR.
>
> This new Table 7 is copied here for convenience:
> | Seed Prompt Generator LLM        | ChatGPT-4o-Latest | GPT-5.1 | Llama-4-Maverick-17B-128E-Instruct |
> |----------------------------------|--------------------|---------|------------------------------------|
> | ASR                              | **76.75**          | 34.50   | 52.25                              |
>
> &nbsp;
>
> ---
>
> ### **Q5. Specify the exact “ChatGPT-4o-Latest” version**
>
> OpenAI does not have date identifiers on its “chat” models such as ChatGPT-4o and ChatGPT-5. Instead, they are simply defined as “chatgpt-4o-latest” and “gpt-5-chat-latest” in the API so the information about the exact date/version is not public.

---

> > ### Author Response · Authors · 2025-11-26
> > **Updated results against Self-Reminder**
> >
> > Dear Reviewer,
> >
> > To address another reviewer’s (Reviewer Se2F’s) concern about JCB’s effectiveness against the Self-Reminder defense, we conducted new experiments in an **adaptive setting**, where the attacker is aware of the defense and instructs the model to operate without Self-Reminder’s restrictions. Under this setting, JCB successfully bypasses the defense, achieving an average ASR of 99.5% across all evaluated models (Table 15, Appendix O in the revised manuscript).
> >
> > This new Table 15 is copied here for convenience:
> > | Models | GPT-3.5 Turbo 1106 | GPT-4o 2024-05-13 | GPT-4o-mini | Llama 3.1 405B | Avg.  |
> > |----------------------|--------------------|-------------------|-------------|-----------------|-------|
> > | JCB ASR          | 100                 | 100                | 98          | 100              | 99.5  |
> >
> >
> > We are informing you here since this is relevant to “Q3. Lack of evaluation under defense methods” in our first response to your review above.

---

> ### Comment · Reviewer_RLgU · 2025-11-28
>
> Thank you for the detailed rebuttal. While I appreciate the clarifications, I still find the novelty of JCB to be limited compared to existing optimization-based jailbreak methods: the scorer-based weighted seed prompt selection and synonym perturbation seem like natural refinements rather than a clearly novel methodology. Moreover, the fact that the non-adaptive JCB cannot bypass defenses such as self-reminder further limits the practical severity of the vulnerability under realistic defensive setups. Due to the incident of openreview system, I can not edit the review score. I would like to adjust the overall score to 4 for this paper.

---

> > ### Author Response · Authors · 2025-12-02
> > **Additional clarifications regarding novelty and resilience against Self-Reminder**
> >
> > Dear Reviewer,
> >
> > Thank you for appreciating our rebuttal and confirming that our clarifications address all but two of your concerns. We address the remaining two concerns next.
> >
> > &nbsp;
> >
> > **On Novelty:**
> >
> > While JCB includes distinct components such as seed-prompt selection and perturbation (which are still different from prior works such as PAIR), **the core methodological novelty lies in cross-behavior learning**: a mechanism that has not appeared in prior jailbreak work, including PAIR, TAP, or other optimization-based attacks. JCB is the first black-box jailbreak method that explicitly leverages behavioral transfer: i.e., using successes from past harmful behaviors to improve query efficiency and success on new behaviors. This mechanism is algorithmically simple, but conceptually novel. None of the 15+ baselines we evaluated incorporate any form of cross-behavior learning, nor do they track behavioral histories to bias future searches.
> >
> > This novelty is not superficial either: empirically, cross-behavior learning enables our method to have 94% fewer queries than PAIR, while still outperforming it by over 12% in ASR. These are not small, incremental gains. Rather, this represents a substantive improvement in efficiency and overall capability under a strict black-box threat model. We believe this also opens a new direction where behavioral generalization can be exploited for achieving high jailbreak success without requiring gradient access, large auxiliary LLMs, or high compute and runtime costs.
> >
> > &nbsp;
> >
> > **On Self-Reminder and Practical Severity:**
> >
> > As noted in our earlier response, our initial rebuttal lacked a complete evaluation against Self-Reminder due to resource constraints. Since then, we conducted a full evaluation in the realistic adaptive-attacker setting. We emphasize that this **adaptive setting is not just appropriate but necessary** because Self-Reminder is public knowledge and its defensive capability should not rely on obscurity (i.e., an attacker not being aware of the defense). As presented in our previous comment here, in this adaptive setting, **JCB successfully bypasses Self-Reminder** across multiple target LLMs, demonstrating its strong effectiveness against this mainstream defense. These results are also included in our updated manuscript (Table 15).
> >
> > &nbsp;
> >
> > In summary, we hope our clarifications make our work’s novelty and resilience against Self-Reminder clear, thereby addressing your remaining two concerns.

---

### Comment · Area_Chair_mJ24 · 2025-11-25

Dear Reviewers,

The authors have submitted their responses to your questions and feedbacks. Please read them and give your comments.

Regards, AC

---

### Author Response · Authors · 2025-12-03
**Summary comment for AC (Part 1 of 2)**

We would like to note that we have attempted to address **all concerns raised by each reviewer in detail** within the discussions below. However, for convenience, we provide here a **high-level summary** of the main concerns from each reviewer and how we resolved them through new experiments, ablations, and clarifications, so the AC can more easily evaluate the submission given the OpenReview disruption.

**Note:** Due to the 5000 character limit for a comment here, we have split this comment in two parts.

&nbsp;

**Summary of our work:** In this work, we propose JCB, a black-box jailbreak method that introduces cross-behavior learning, a novel mechanism that reuses successful jailbreak prompts from one harmful behavior to improve success and efficiency on others. Combined with lightweight part-of-speech-preserving synonym perturbation, JCB avoids expensive auxiliary LLMs while remaining semantically coherent. Across 20+ open- and closed-source models, JCB achieves state-of-the-art ASR with up to 94% fewer queries than prior black-box methods. It outperforms all SOTA baselines, shows strong cross-model transfer, and bypasses popular defenses such as Self-Reminder and perplexity-based filtering. Ablations and mechanistic analysis confirm that cross-behavior learning is essential to these gains.


&nbsp;

Next, we summarize the reviewers’ concerns and how we addressed them.

&nbsp;

---

### **Reviewer RLgU**

**Concerns:** novelty of JCB relative to PAIR/TAP; missing baseline comparisons on SOTA models; lack of evaluation under defenses (especially Self-Reminder).

**How we addressed them:**

* We clarified that JCB introduces a **novel cross-behavior learning mechanism**, absent in prior work, enabling **94% query-efficiency gains** over PAIR/TAP. Extensive experiments along with ablations (Appendix K) empirically confirm its importance.
* We added **new SOTA baseline comparisons** (PAIR, TAP, BOOST, LLMFuzzer, and FlipAttack) on SOTA models such as GPT-3.5, GPT-4o, Llama-3.1-405B, etc. where JCB outperforms all baselines (Appendix I).
* We added **comprehensive defense evaluations**, including an evaluation showing that JCB **bypasses Self-Reminder with 99.5% ASR** as well as another perplexity-based defense (Appendix O).

Note that Reviewer RLgU’s last comment (i.e., before the OpenReview incident) indicated that our rebuttal addressed all but two concerns, so our last comment in that thread ([link](https://openreview.net/forum?id=OkjB6PWJEA&noteId=Lds7iM5vgT)) addresses those two remaining concerns.

&nbsp;

---

### **Reviewer DUEw**

**Concerns:** missing ablations; unclear source of gains; limited diversity; fairness of comparisons; baseline correctness.

**How we addressed them:**

* We added an **ablation study** (Appendix K) separating the contributions of jailbreak themes and our selection and perturbation components. This clarifies the source of performance gains.
* We contextualized the use of two high-performing jailbreak themes and showed via ablations that **all components contribute**, not only priors.
* We clarified that while the stylistic diversity of prompts is narrower, JCB’s cross-behavior learning provides **a new and impactful algorithmic contribution** that uncovers systematic behavioral vulnerabilities across models.
* We clarified fairness with the fact that JCB simply builds on established findings, a standard and essential scientific practice, rather than unfair advantage.
* We confirmed that **all main baseline results** come from the unified HarmBench framework for apples-to-apples evaluation.

Note that Reviewer DUEw’s last comment (i.e., before the OpenReview incident) indicated that our rebuttal addressed all concerns except that of our contributions to LLM safety, so our last comment in that thread ([link](https://openreview.net/forum?id=OkjB6PWJEA&noteId=BM1vykVYbN)) addresses that remaining concern.

&nbsp;

---

=======continued in the next comment=======

---

> ### Author Response · Authors · 2025-12-03
> **Summary comment for AC (Part 2 of 2)**
>
> =======continued from the previous comment=======
>
> &nbsp;
>
> ---
>
> ### **Reviewer Se2F**
>
> **Concerns:** lack of validation against defenses; insufficient theory; questions about priors and ablations.
>
> **How we addressed them:**
>
> * We added **defense evaluations**, including perplexity-based filtering (fully bypassed) and an **adaptive Self-Reminder evaluation**, where JCB achieves **99.5% ASR** (Appendix O).
> * We added a **new mechanistic analysis** (Appendix L) showing that JCB shifts activations away from harmful regions and toward harmless ones, helping explain our work’s underlying mechanisms.
> * We added ablations isolating the effect of priors and showing that **cross-behavior learning and perturbation are essential**, independent contributors (Appendix K).
>
> Note that Reviewer Se2F’s last comment (i.e., before the OpenReview incident) indicated that our rebuttal addressed all concerns except that of validation against defenses, so our last comment in that thread ([link](https://openreview.net/forum?id=OkjB6PWJEA&noteId=Fi4Ejgykdp)) addresses that remaining concern.
>
> &nbsp;
>
> ---
>
> ### **Reviewer mLEf**
>
> **Concerns:** reproducibility statement; novelty relative to LLMFuzzer; missing SOTA baselines; performance under Self-Reminder.
>
> **How we addressed them:**
>
> * We added a complete **reproducibility statement** (Section 7).
> * We clarified the novelty: **explicit cross-behavior learning** and **lightweight POS-preserving perturbation** (neither of which appears in LLMFuzzer) and provided empirical evidence showing JCB’s large improvements.
> * We added new **BOOST, LLMFuzzer, and FlipAttack** comparisons (Appendix I) which show that JCB consistently outperforms them.
> * We added a **new evaluation against Self-Reminder defense**, showcasing that JCB easily bypasses the defense with an average ASR of 99.5%.
>
> Note that Reviewer mLEf’s last comment (i.e., before the OpenReview incident) indicated that our rebuttal addressed all their concerns and stated that they **are open to acceptance if other reviewers’ concerns have been addressed** ([link](https://openreview.net/forum?id=OkjB6PWJEA&noteId=JUJOjhVsRJ)).
>
> &nbsp;
>
> ---
>
> ### **Closing**
>
> We thank the AC for their time and effort in evaluating this submission, especially under the unexpected OpenReview constraints. We hope this concise summary helps contextualize the detailed discussions and the substantial additional experiments, analyses, and clarifications we have provided.
>
> If any further information would be helpful, we are happy to provide it.

---

### Meta-Review · Area_Chair_kwHK · 2025-12-10

**Summary:**

This paper proposes a black-box Jailbreak method with Cross-Behavior attacks that can automatically and efficiently find successful jailbreak prompts. JCB does not rely on time- and/or cost-intensive calls to auxiliary LLMs to discover/optimize the jailbreak prompts, making it highly efficient and scalable.

**Reviewer Concerns:**

All the reviewer find that the novelty of JCB is rather limited compared to existing optimization-based jailbreak methods. Because the proposed method of selecting seed prompts based on prior attack successes and performing synonym-based perturbations is quite similar to existing approaches (e.g., PAIR [1] using an LLM to rewrite prompt seeds).

**Reviewer Scores:**

The novelty of JCB is rather limited.

---

### Decision · Program_Chairs · 2026-01-26

Reject